# A hygroscopic nano-membrane coating achieves efficient vapor-fed photocatalytic water splitting

Takuya Suguro[1], Fuminao Kishimoto [1], Nobuko Kariya[2], Tsuyoshi Fukui[2], Mamiko Nakabayashi [3], Naoya Shibata [3], Tsuyoshi Takata[4], Kazunari Domen [4,5] & Kazuhiro Takanabe [1] ✉

Efficient water vapor splitting opens a new strategy to develop scalable and corrosion-free solar-energy-harvesting systems. This study demonstrates highly efficient overall water splitting under vapor feeding using Al-doped $SrTiO_3$ ($SrTiO_3$:Al)-based photocatalyst decorated homogeneously with nano-membrane $TiO_x$ or $TaO_x$ thin layers (<3 nm). Here, we show the hygroscopic nature of the metal (hydr)oxide layer provides liquid water reaction environment under vapor, thus achieving an AQY of $54 \pm 4\%$, which is comparable to a liquid reaction. $TiO_x$ coated, CoOOH/Rh loaded $SrTiO_3$:Al photocatalyst works for over 100 h, under high pressure (0.3 MPa), and with no problems using simulated seawater as the water vapor supply source. This vapor feeding concept is innovative as a high-pressure-tolerant photoreactor and may have value for large-scale applications. It allows uniform distribution of the water reactant into the reactor system without the potential risk of removing photocatalyst powders and eluting some dissolved ions from the reactor.

Photocatalytic water splitting is one of the most important reactions in a hydrogen-based society[1–3]. Water splitting from a particulate semiconductor photocatalyst in the absence of an external bias is potentially scalable and technologically feasible[4]. A recent field test was conducted on $100 m^2$ of particulate Al-doped $SrTiO_3$ ($SrTiO_3$:Al) photocatalyst sheets where the maximum solar to hydrogen (STH) efficiency reached 0.76%[5]. On the laboratory-scale, a STH efficiency of 1.1% was recorded by the dual-photocatalyst sheets consisting of La and Rh co-doped $SrTiO_3$ ($SrTiO_3$:La, Rh) and Mo-doped $BiVO_4$ ($BiVO_4$:Mo) powders immobilized on a gold (Au) layer[6]. These photocatalyst powders were immobilized on a sheet, and the liquid water was flown over it. The resulting gases were collected in a sealed reactor. The STH efficiency of 10% is often the benchmark target for green water splitting technology for commercial feasibility[7]; further material and system developments are required to improve the efficiency of photocatalytic systems for practical use.

Particulate photocatalysts typically have an interface with liquid water, which reduces their choice of materials due to photocorrosion and dissolution of semiconductors[8–10] and cocatalysts[11]. Flowing viscous water into the reactor can continuously elute the photocatalyst powder itself or the dissolved ions[12]. Meter-scale photoreactors must also be designed for uniform distribution with the correct angle of the photocatalyst panels. These must be tolerant to volume changes in liquid water upon temperature changes during day and night[5,12]. Although water vapor splitting systems can mitigate the above-mentioned issues of the conventional liquid phase system, the water vapor splitting system has not been intensely investigated because of lack of strategies to archive high efficiencies[13].

[1]Department of Chemical System Engineering, School of Engineering, The University of Tokyo, 7-3-1 Hongo, Bunkyo-ku, Tokyo 113-8656, Japan. [2]Science & Innovation Center, Mitsubishi Chemical Corporation, 1000 Kamoshida-cho, Aoba-ku, Yokohama, Kanagawa 227-8502, Japan. [3]Institute of Engineering Innovation, School of Engineering, The University of Tokyo, 7-3-1 Hongo, Bunkyo-ku, Tokyo 113-8656, Japan. [4]Research Initiative for Supra-Materials (RISM), Shinshu University, 4-17-1 Wakasato, Nagano 380-8553, Japan. [5]Office of University Professors, The University of Tokyo, 7-3-1 Hongo, Bunkyo-ku, Tokyo 113-8656, Japan. ✉e-mail: takanabe@chemsys.t.u-tokyo.ac.jp

The first demonstrations of photocatalytic overall water splitting were by Schrauzer et al. and Domen et al. and used water vapor feeding[14,15]. In 2011, Dionigi et al. reported water vapor splitting on $RhCrO_x$ loaded GaN:ZnO photocatalysts using a μ–reactor to comprehensively analyze the reaction kinetics of water splitting reaction[16]. They proposed that the proton conductivity in the adsorbed water layers between oxidation and reduction reaction center is crucial for efficient water splitting under vapor feeding. To date, the particulate $RhCrO_x/SrTiO_3$:Al/CoOOH photocatalyst exhibits only 2% AQY at 365 nm-LED light (14.3 mW cm$^{-2}$) with a relative humidity of 0.58 at 303 K in contrast to an AQY of >50% in liquid water[17].

The reason for the low efficiency under vapor feeding resides with at least one of the following six elementary steps during photocatalytic reaction: (1) photon absorption by a photocatalyst, (2) exciton separation, (3) carrier diffusion, (4) carrier transport, (5) electrocatalytic reactions, and (6) mass transfer of reactants and products[18]. The $Rh/Cr_2O_3$ and CoOOH co-deposited $SrTiO_3$:Al photocatalyst exhibits an AQY of almost unity in liquid water under UV light irradiation[19], thus indicating that there are no losses in initial 4 elementary steps. A lower AQY of $SrTiO_3$:Al photocatalyst under vapor feeding is thus due to the losses of efficiencies in steps (5) and (6) associated with the reactants. To complete the electronic circuit on the photocatalyst surface, the reduction and oxidation sites present on the same surface must close the electric circuit as discussed in the literature[18]. Several attempts at physical mixing of particulate photocatalysts with hydrophilic reagents such as $MoS_x$[19], hydrogels[20], and NaOH[21] have been reported, but they all suffered from the low gas diffusion through the layer or strong corrosion toward the photocatalyst materials.

Here, efficient water splitting under water vapor feeding with as high of a AQY as that in liquid water was demonstrated using particulate $SrTiO_3$:Al photocatalyst with an amorphous inorganic metal hydroxides ultrathin layer (Fig. 1); the uniform coating of the hydroxides was achieved via the photodeposition method[22]. The high conductivity of the metal hydroxides was comparable to a typical proton-conductive material such as Nafion[23,24]; thus, the coated thin-layer hydroxides had a sufficient amount of adsorbed water and can close the electric circuit with high conductivity between the $H_2$ evolution reaction site and the $O_2$ evolution site. This vapor feeding allows very different perspectives for large-scale photoreactor design: Versus liquid feeding, a uniform distribution of the water reactant is easily achievable via gas introduction, and the high-pressure photocatalytic reactor is more easily established.

## Results

### Water splitting under water vapor feeding

A photograph and a schematic of the photocatalytic reactor with continuous flow of water vapor are shown in Supplementary Fig. 1. The photocatalytic test was performed using the flow reactor. The Ar inlet gas was passed through a water bubbler and then supplied into the photocatalyst reactor made of stainless steel (SUS304). The photocatalyst sheet is where the particulate photocatalysts were immobilized on a glass plate as shown in Fig. 2a. The sheet was placed in the stainless reactor and irradiated with a light-emitting diode (LED) at 370 nm through a glass window at the top of the stainless reactor. A detailed spectrum of the light source is shown in Supplementary Fig. 2. The hygrometer was introduced between the water bubbler and the stainless reactor to record the relative humidity (RH) in the feed gas. The relative humidity was precisely controlled by mixing the wet Ar gas (RH = 1.0) with dry Ar gas (RH = 0). The water bubbler, the hygrometer, and the stainless reactor were placed in a heating oven to maintain the temperature and the water partial pressure. The outlet gas was analyzed by GC−TCD to quantify the generated $H_2$ and $O_2$ gas.

X-ray diffraction (XRD) patterns (Supplementary Fig. 3) show that the $SrTiO_3$:Al photocatalysts used here were single phase; crystalline $TiO_2$ phase was not observed even after $TiO_x$ coating. The light absorption edge of the $SrTiO_3$:Al-based catalysts (Supplementary Fig. 4) was located near 380 nm, which is consistent with the previous reports[12,25,26]. Figure 2b shows the $H_2$ and $O_2$ production rate as a function of time on stream under various irradiated light intensities using CoOOH and Rh co-loaded $SrTiO_3$:Al photocatalyst coated with $TiO_x$. The feed gas was Ar with saturated water vapor at 24 °C. During the demonstration, the ratio of $H_2$ and $O_2$ evolution was maintained at nearly 1:2, thus showing that the electrons and holes were consumed by the water splitting reaction; no other side reaction occurred. When the light intensity was lower than 25 mW cm$^{-2}$, the gas evolution rate was stable with the time on stream. Importantly, the rate gradually decreased under higher intensity light irradiation (>59 mW cm$^{-2}$). The cause of decrease in AQY as the function of light intensity as well as decrease in formation rate with time will be discussed in the characterization of oxide thin-layer section. The experiments described hereafter were performed under a light intensity of approximately 5 mW cm$^{-2}$, which corresponds to the actual intensity of UV light in AM 1.5G solar light.

Figure 2b also shows that the $H_2$ evolution rate reached to ~150 μmol h$^{-1}$ per 1 cm$^2$ photocatalyst sheet, demonstrating the applicability of the $TiO_x$ coating for improved photocatalysts where solar-to-hydrogen (STH) efficiency of 10% (corresponding to a hydrogen production rate of 150 μmol h$^{-1}$ per 1 cm$^2$) is achievable. This demonstration suggested that this $H_2$ production system is not limited by water vapor supply even if the $H_2$ evolution rate is corresponding to the rate of 10% STH. The partial pressure of $H_2$ in the outlet gas was estimated to be 0.56 kPa.

The $H_2$ evolution rate of $SrTiO_3$:Al-based catalysts loaded with various cocatalysts and oxides is summarized in Fig. 2c where the incident light intensity was fixed as 5.1 mW cm$^{-2}$. The temperature was 24 °C, and the relative humidity in the Ar feed gas was near unity. The $H_2$ and $O_2$ gas evolution rates of each catalyst as a function of time on stream are shown in Supplementary Figs. 5–10. The apparent quantum yield of $SrTiO_3$:Al loaded with $RhCrO_x$ via the impregnation method in the current demonstration was higher than the AQY reported by Shearer et al. (~2%) due to a higher light intensity (14.27 mW cm$^{-2}$) and lower relative humidity (58%) than the previous reports[17]. Versus CoOOH/Rh or CoOOH/$Cr_2O_3$/Rh loaded $SrTiO_3$:Al photocatalysts, these photocatalysts with $TiO_x$ coating obviously enhanced the AQY. The $TiO_x$ coated CoOOH/$Cr_2O_3$/Rh loaded $SrTiO_3$:Al photocatalyst exhibited a AQY as high as ~60%, which is comparable to the reported AQY value of the CoOOH/$Cr_2O_3$/Rh loaded $SrTiO_3$:Al photocatalyst in liquid water under 370 nm light[19]. Enhanced AQY was not observed in the case of a physical mixture of CoOOH/Rh loaded $SrTiO_3$:Al and $TiO_2$ nanoparticles (P25).

The $TiO_x$ coated CoOOH/Rh loaded $SrTiO_3$:Al photocatalyst exhibited an AQY comparable to that of $TiO_x$ coated CoOOH/$Cr_2O_3$/Rh loaded $SrTiO_3$:Al photocatalyst. The $Cr_2O_3$ co-loaded with Rh cocatalyst has been reported for improved overall apparent quantum yield of

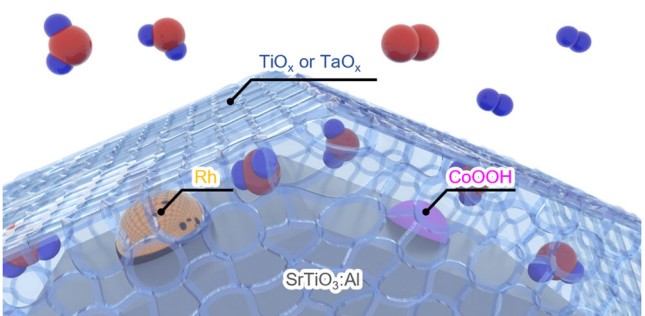

**Fig. 1 | Schematic illustration of amorphous metal oxide coated, CoOOH/Rh loaded $SrTiO_3$:Al.** Red: O atom, blue: H atom.

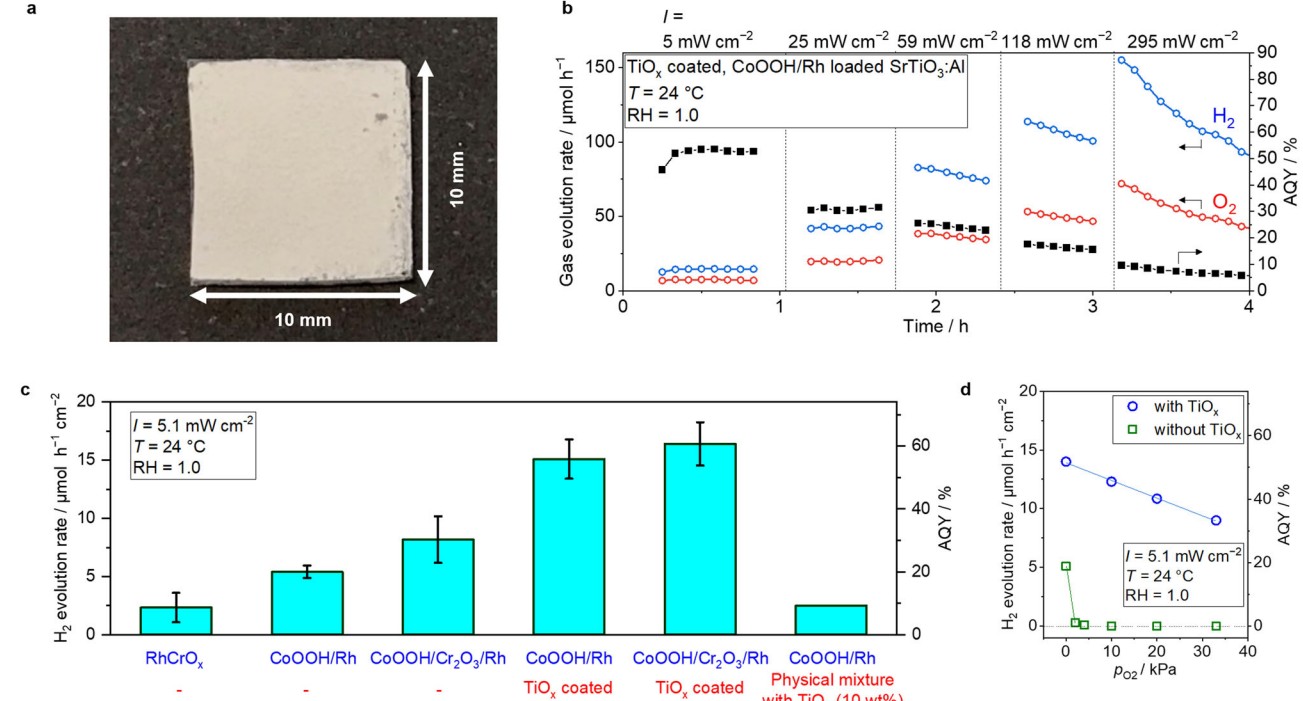

**Fig. 2 | SrTiO₃:Al-based photocatalysts for overall water splitting under vapor feeding. a** Photographic images of a photocatalyst sheet (1 × 1 cm²), which consists of SrTiO₃:Al-based particulate photocatalysts immobilized on a frosted glass substrate. **b** Time course measurement of H₂ and O₂ gas evolution rate using CoOOH/Rh loaded SrTiO₃:Al coated with TiO$_x$ under various light intensity. The fed gas was Ar with saturated water vapor at 24 °C (10 mL min⁻¹, $p_{H2O}$ = 2.9 kPa). **c** H₂ evolution rate of various cocatalysts loaded SrTiO₃:Al irradiated with 370 nm LED (5.1 mW cm⁻²) under saturated water vapor balanced with Ar at 24 °C (10 mL min⁻¹, $p_{H2O}$ = 2.9 kPa). Error bars are standard error from other samples (n = 3–11). **d** Dependence of H₂ evolution rate of CoOOH/Rh loaded SrTiO₃:Al coated with TiO$_x$ (blue circle) and without TiO$_x$ (green square) under various O₂ partial pressure ranged from 0 to 33 kPa under 370 nm LED (5.1 mW cm⁻²).

water splitting in liquid water by preventing the reverse reaction of water generation from H₂ and O₂[25,27]. However, because of the possibility of harmful Cr leaching, detailed characteristics of Cr-free TiO$_x$ coated CoOOH/Rh loaded SrTiO₃:Al photocatalyst were investigated. The AQY of water splitting under vapor feeding using the photocatalysts with or without Cr₂O₃ was almost unchanged, as shown in Fig. 2c. Indeed, it has been previously demonstrated that TiO$_x$ layer acting as an H₂-selective nano-membrane can prevent the reverse water-forming reaction in liquid water[6]. To demonstrate the dynamics of reverse water-forming reaction from H₂ and O₂, photocatalytic performance in the presence of O₂ was tested as shown in Fig. 2d. The apparent H₂ evolution rate immediately reached zero in the presence of 4% O₂ using CoOOH/Rh loaded SrTiO₃:Al photocatalyst without TiO$_x$ layer. On the other hand, in the case of CoOOH/Rh loaded SrTiO₃:Al photocatalyst with TiO$_x$ layer, the AQY was maintained above 40% even in the presence of 20% O₂. Therefore, TiO$_x$ layer has the role of preventing the reverse water-forming reaction even under O₂ presence.

### Characterization of oxide thin layer

Figure 3a shows a bright field scanning transmission electron microscopy (STEM) image of TiO$_x$ coated CoOOH/Rh loaded SrTiO₃:Al photocatalyst. An amorphous uniform surface layer with a thickness of about 3 nm was observed as a slightly brighter region than the crystalline core region. Overlayed energy dispersive X-ray spectroscopy (EDS) map of the corresponding area (Fig. 3b) shows that the core region consisted of Sr and Ti elements, while the surface layer only contained Ti. Therefore, the core region and the uniform surface layer can be attributed to SrTiO₃:Al photocatalyst and TiO$_x$ coated layer, respectively. The EDS line scan in Supplementary Fig. 11 shows the existence of Rh nanoparticles with a Ti-rich surface. X-ray photoelectron spectroscopy (XPS) results shows the drastic attenuation of the Sr

3*d* peak after TiO$_x$ coating while the Ti 2*p* peak was maintained also demonstrating the uniform TiO$_x$ coating on the surface of SrTiO₃:Al photocatalyst (Supplementary Fig. 12). The particle size of the photocatalyst was approximately 500 nm as shown in Supplementary Fig. 13.

The water adsorption isotherm in Fig. 3c demonstrated that TiO$_x$ coating on the CoOOH/Rh loaded SrTiO₃:Al photocatalyst drastically improved the water adsorption amount on the photocatalyst at the specific relative humidity. The amount of water adsorbed significantly increased after TiO$_x$ coating from 0.12 mmol g⁻¹ to 1.39 mmol g⁻¹ at a relative humidity of 0.8, i.e., the adsorbed water amount on CoOOH/Rh loaded SrTiO₃:Al was increased 10-fold by TiO$_x$ coating. On the other hand, CoOOH or Cr₂O₃ loading did not exhibit high water adsorption ability (Supplementary Figs. 14 or 15, respectively) compared to TiO$_x$ coating. The isotherm of TiO$_x$ coated CoOOH/Rh loaded SrTiO₃:Al showed type-II isotherm, and the liquid-like water should exist on the photocatalyst surface under relative humidity values higher than 0.6. Hence, the photocatalyst can exhibit water splitting performance under vapor feeding similar to when the photocatalyst is immersed in liquid water. Therefore, the role of the coated TiO$_x$ is mitigation of the loss in the elementary reaction step (6) as well as mass transfer of reactants during the overall photocatalytic reaction.

The relative humidity in the feed gas influences the photocatalytic gas production rate based on the water adsorption isotherm. The H₂ evolution rate of TiO$_x$ coated CoOOH/Rh loaded SrTiO₃:Al photocatalyst under various humidity and temperature values are summarized in Fig. 3d. Supplementary Fig. 16 shows the time on stream H₂ and O₂ evolution rate where the relative humidity in the feed gas varied: 1.0, 0.8, 0.6, 0.4, 0.2, 0.1 and finally back to 1.0 at temperatures of 24, 50, and 80 °C. At 24 and 50 °C, the apparent quantum yield did not depend on the reaction temperature but rather decreased with decreasing relative humidity. At a relative humidity above 0.6, the TiO$_x$ coated CoOOH/Rh loaded SrTiO₃:Al photocatalyst exhibited higher

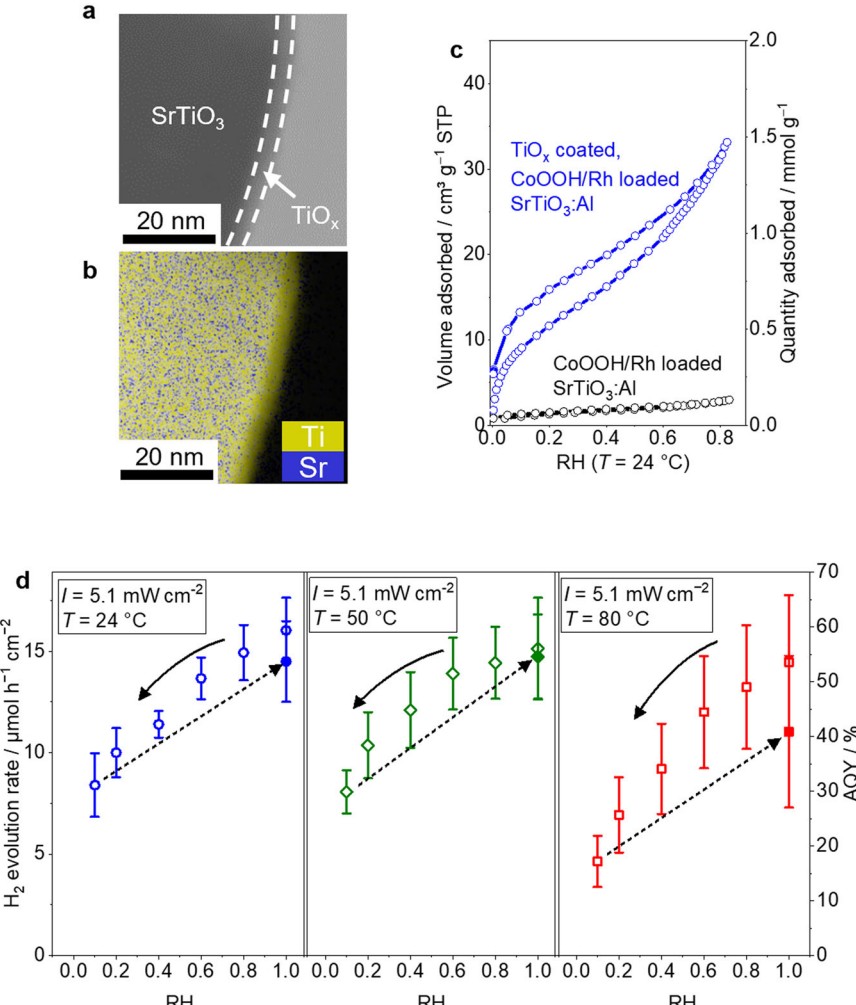

**Fig. 3 | The role of the coated TiO$_x$ on CoOOH/Rh loaded SrTiO$_3$:Al. a** Bright field STEM image of CoOOH/Rh loaded SrTiO$_3$:Al coated with TiO$_x$ and **b** its corresponding overlayed EDS mapping of Ti and Sr. **c** Water adsorption isotherm of CoOOH/Rh loaded SrTiO$_3$:Al coated with or without TiO$_x$. **d** Dependence of H$_2$ evolution rate of CoOOH/Rh loaded SrTiO$_3$:Al coated with TiO$_x$ on relative humidity at 24 °C (blue circle), 50 °C (green diamond), and 80 °C (red square). Error bars are standard error from other samples ($n$ = 3).

AQY (>50%), thus suggesting that the water on the photocatalyst surface demonstrated by water adsorption isotherm is crucial for efficient photocatalytic water splitting under vapor feeding. A relatively lower apparent quantum efficiency was observed at 80 °C, and the H$_2$ and O$_2$ evolution rate gradually decreased with reaction time (Supplementary Fig. 16). Unlike the case at 24 and 50 °C, the reaction rate at 80 °C did not recover even when the humidity returned from 0.1 to 1, thus suggesting that the TiO$_x$ covering layer was irreversibly restructured.

TaO$_x$ can also be decorated as a thin layer on the CoOOH/Rh loaded SrTiO$_3$:Al, exhibiting the high water splitting performance under water vapor feeding as well as TiO$_x$ coated CoOOH/Rh loaded SrTiO$_3$:Al photocatalyst. Figure 4a shows a dark field scanning transmission electron microscope (STEM) image of TaO$_x$ coated CoOOH/Rh loaded SrTiO$_3$:Al photocatalyst and Fig. 4b shows its corresponding EDS mapping of Ta. An amorphous uniform surface layer with a thickness of 2–3 nm was also observed as with the photocatalyst coated with TiO$_x$. Figure 4c shows the relative humidity dependence of the AQY of TaO$_x$-coated photocatalyst. The dependence was similar to the TiO$_x$ coated CoOOH/Rh loaded SrTiO$_3$:Al photocatalyst. The TaO$_x$ coated CoOOH/Rh loaded SrTiO$_3$:Al photocatalyst also exhibited a high AQY (>50%) under water vapor feeding with a relative humidity of 1.0 (Fig. 4d), which was comparable to TiO$_x$ coated CoOOH/Rh loaded SrTiO$_3$:Al photocatalyst.

Hereinafter, the origin of the decrease of water splitting rate under high-temperature condition, as shown in Fig. 3d will be discussed. To investigate the structural change of the covered oxide layer after high-temperature treatment, TaO$_x$ coated photocatalyst was used instead of TiO$_x$ to avoid the overlapped signal from SrTiO$_3$. Figure 4d shows a drastic decrease of the apparent quantum efficiency after heat treatment of the photocatalyst at 250 °C in a convection flow oven. Figure 4e shows Fourier transformed Ta L$_{III}$ edge extended X-ray absorption fine structure (EXAFS) profiles of as-synthesized (60 °C dried) TaO$_x$ coated CoOOH/Rh loaded SrTiO$_3$:Al photocatalyst and after 250 °C heat treatment. EXAFS oscillations before Fourier transformation are shown in Supplementary Fig. 17. The peaks at 1.4 Å in the radial distribution functions, attributed to the neighboring oxygen shell[28], were deconvoluted into two peaks assigned as Ta-O$_1$ and Ta-O$_2$ based on the crystal structure of orthorhombic Ta$_2$O$_5$ (Supplementary Fig. 18). Table 1 summarizes the fitting results of Debye-Waller factor ($\sigma^2$) and the interatomic distance ($R$). Because these measurements were performed at a constant temperature, the Debye-Waller factor ($\sigma^2$) corresponds to the statistical degree of disorder in atomic positions with respect to the ideal crystal structure of Ta$_2$O$_5$. Accompanied by the decrease of $\sigma^2$ of both Ta-O$_1$ and Ta-O$_2$ peaks after the treatment at 250 °C, the interatomic distance ($R$) of Ta and O became slightly shorter. These results suggested that, as a statistically significant difference, TaO$_x$ layer

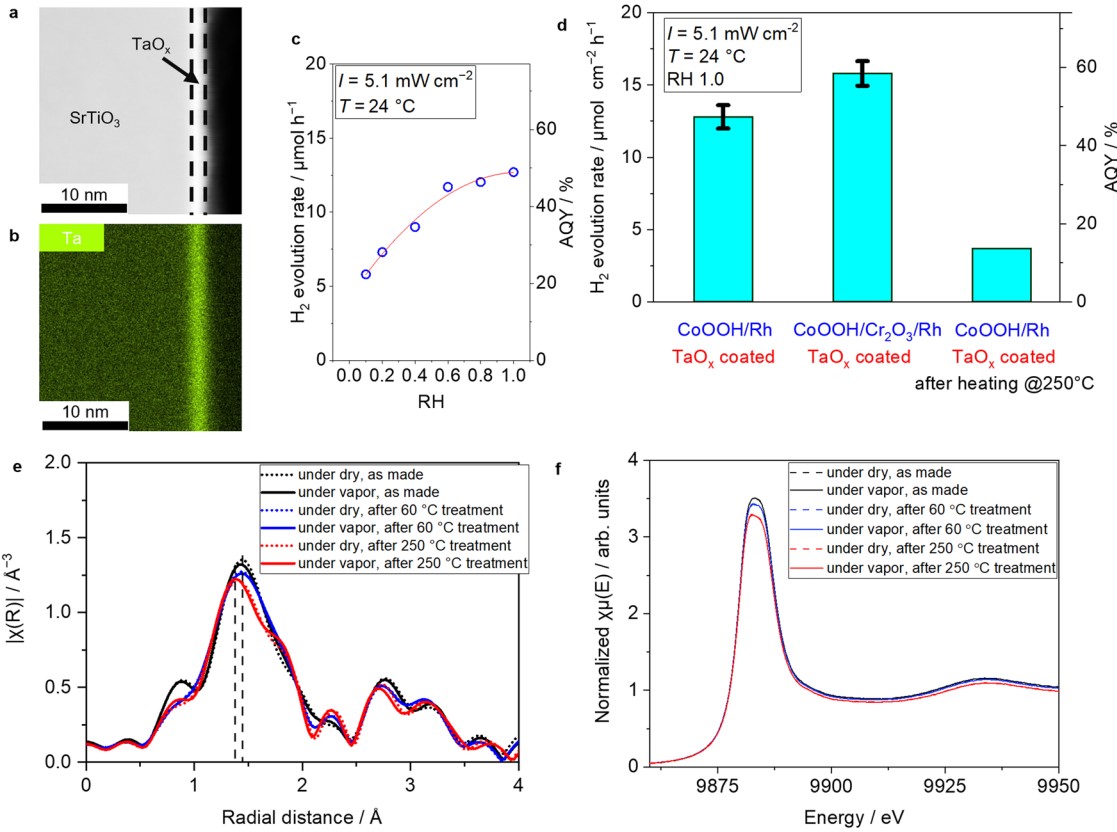

**Fig. 4 | Thermal deactivation of TaO$_x$ coated SrTiO$_3$:Al photocatalysts. a** Dark field TEM image of CoOOH/Rh loaded SrTiO$_3$:Al coated with TaO$_x$ and **b** its corresponding EDS mapping of Ta. **c** H$_2$ evolution rate of TaO$_x$ coated SrTiO$_3$:Al type photocatalysts before and after thermal treatment and regeneration process. These demonstrations were under saturated water vapor balanced with Ar at 24 °C (10 mL min$^{-1}$, $p_{H2O}$ = 2.9 kPa). Light source: 370 LED (5.1 mW cm$^{-2}$). Error bars are standard error from other samples ($n$ = 3). **d** Dependence of H$_2$ evolution rate of CoOOH/Rh loaded SrTiO$_3$:Al coated with TaO$_x$ on relative humidity at 24 °C. Light source: 370 nm LED (5.1 mW cm$^{-2}$). **e** Fourier transformed Ta L$_{III}$ EXAFS patterns of CoOOH/Rh loaded SrTiO$_3$:Al coated with TaO$_x$ recorded at 24 °C under dried condition (dashed) and (solid) after thermal treatment, and **f** corresponding XANES region of Ta L$_{III}$ XAFS spectra of TaO$_x$ coated, CoOOH/Rh loaded SrTiO$_3$:Al.

should become denser and comparatively ordered crystal structure after the treatment at 250 °C. It is worth noting that there is no significant change in the X-ray absorption near edge structure (XANES) region before and after heat treatment (Fig. 4f).

TG–DTA measurement (Supplementary Figs. 19, 20) also shows that the weight loss by dehydration occurs starting at 45 °C, consistent with the structural change measured by EXAFS. These results suggested that an irreversible structural change of the TiO$_x$ or TaO$_x$ layer was induced by loss of water molecules from the layer, resulting in reduced water adsorption and lower H$_2$ production rate. The decrease in AQY as increasing light intensity and the reaction time shown in Fig. 2d can be due to the loss of water molecules in the TiO$_x$ or TaO$_x$ layer either by temperature rise or higher consumption rate of water molecules than their supply by water splitting reaction. Diffuse reflectance infrared spectroscopy measurement, as shown in Supplementary Fig. 21 supports the loss of OH species from the TiO$_x$ layer under the 370 nm LED irradiation with high light intensity.

## Water splitting under practical conditions

Figure 5a shows the results of the long-term stability test under AM 1.5G simulated sunlight (~100 mW cm$^2$) using TiO$_x$ coated CoOOH/Rh loaded SrTiO$_3$:Al photocatalyst sheet immobilized on 2 × 2 cm$^2$ glass plate. Under vapor feeding, H$_2$ and O$_2$ were evolved in a stoichiometric ratio (1:2), and the average solar to hydrogen (STH) energy conversion efficiency was at around 0.4%, which is comparable to a field test result conducted on a 100 m$^2$ area of particulate CoOOH/Cr$_2$O$_3$/Rh loaded SrTiO$_3$:Al photocatalyst sheets using liquid

water feeding[6]. To avoid heat accumulation on a photocatalyst sheet under simulated sunlight irradiation, a cooling plate driven by a Peltier element (ASONE, SCP-125; set temperature = 25 °C) was installed under the stainless photocatalyst reactor in the current demonstration. During the 100-hour durability test, the generated H$_2$ was over 2 mmol while the TG-DTA shown in Supplementary Fig. 21 revealed that the amount of adsorbed water on as synthesized TiO$_x$ coated CoOOH/Rh loaded SrTiO$_3$:Al photocatalyst sheet (2 × 2 cm$^2$) was approximately 56 μmol. These results confirmed that adsorbed water enriched by the TiO$_x$ coating improved the photocatalytic efficiency.

The TiO$_x$ coated CoOOH/Rh loaded SrTiO$_3$:Al photocatalyst exhibited similar STH under vapor feeding from 3 wt% NaCl$_{aq.}$ compared to water vapor from pure water, thus demonstrating that seawater can be utilized as the vapor source (Fig. 5b). Previously, seawater for liquid phase photocatalytic water splitting was demonstrated for a composite particulate photocatalyst consisting of carbon-nanodots and graphitic carbon nitride (C$_3$N$_4$) where the STH drastically dropped from 2.0% to 0.46%[29]. In contrast, the STH remained near 6.0% within 50 h of long operation using a vapor feeding from seawater in an integrated system of photovoltaic arrays with a water electrolyzer; meanwhile, a drop of STH from 6.5% to 0.5% was seen when using liquid seawater feeding[30].

Finally, we demonstrated H$_2$ and O$_2$ evolution using TiO$_x$ coated CoOOH/Rh loaded SrTiO$_3$:Al photocatalyst sheets under pressurized condition via a tailor-made acrylic reactor with an internal volume of 200 cm$^3$ (Fig. 5c). The reactor was filled with ultrapure water, and a photocatalyst sheet (4 × 4 cm$^2$) immobilizing TiO$_x$ coated CoOOH/Rh

SrTiO$_3$:Al was placed on a pedestal floated in the water. The water vapor was supplied to the photocatalyst sheet by the nitrogen gas flow through the water. Figure 5d shows H$_2$ evolution rate under various total gas pressures controlled by the back pressure regulator installed downstream. There was no significant decrease in the rate of water splitting under vapor feeding at 0.3 MPa versus that at 0.1 MPa.

## Discussion

An apparent quantum yield of 54 ± 4% with 370 nm LED light for water splitting reaction under vapor feeding was achieved using Cr-free CoOOH and Rh co-deposited SrTiO$_3$:Al with uniformly coated metal (Ti or Ta) (hydr)oxide thin layers. This surface nano-membrane coating captures sufficient amount of water vapor, thus generating the liquid environment at the surface of SrTiO$_3$:Al. The relative humidity pins the photocatalytic performance at each reaction temperature. Consequently, a comparable AQY in water can be achieved under vapor feeding. The solar-to-hydrogen efficiency was maintained at around 0.4% for 100 h of long operation, and seawater as the water vapor source did not affect the performance. These data suggest a new design for efficient, durable, and scalable solar H$_2$ production systems using the ambient humidity in a seawater feedstock.

**Table 1 | Debye–Waller factor ($\sigma^2$) and interatomic distance ($R$) from FEFF fitting of EXAFS shown in Supplementary Fig. 18, where Ta-O$_1$ represents scattering concerning the nearest oxygen and Ta-O$_2$ concerning the second nearest oxygen**

|  |  | $\sigma^2$ | $R$ |
|---|---|---|---|
| Under vapor, after drying at 60 °C | Ta-O$_1$ | 0.011 | 1.91 |
|  | Ta-O$_2$ | 0.0059 | 2.12 |
| Under vapor, after drying at 250 °C | Ta-O$_1$ | 0.0087 | 1.87 |
|  | Ta-O$_2$ | 0.0036 | 2.11 |

Orthorhombic Ta$_2$O$_5$ crystal structure (Pmmm) was used as a reference structure.

## Methods

### Synthesis of Al-doped SrTiO$_3$

The SrTiO$_3$ doped with Al (SrTiO$_3$:Al) was synthesized via a previously reported method[25]. SrTiO$_3$ (FUJIFILM Wako Pure Chemical Corporation) was mixed with SrCl$_2$ (Kanto Chemical Co., Inc.) and Al$_2$O$_3$ (MilliporeSigma) by grinding in an agate mortar at a molar ratio of 10:0.02:1. The resulting mixture was moved into an alumina crucible, heated at 1150 °C for 10 h, and then allowed to cool to room temperature. The product was separated from unreacted SrCl$_2$, Al$_2$O$_3$, and other impurities, and the resulting powder was then washed with massive amounts of distilled water using ultrasonication until the Cl$^-$ ions were not detected in the supernatant. The powder was obtained by suction filtration and dried in an oven at 60 °C overnight to obtain a white powder (SrTiO$_3$:Al).

### Cocatalyst loading

The rhodium and chromium mixed oxide (RhCrO$_x$) was loaded on SrTiO$_3$:Al via an impregnation method following the literature[13]. The SrTiO$_3$:Al (0.3 g) was dispersed in ultrapure water (3.34 mL) and then 0.1 M Na$_3$RhCl$_6$ (Mitsuwa Chemicals Co., Ltd.) aqueous solution (50 μL) and 0.05 M Cr(NO$_3$)$_3$ (MilliporeSigma) aqueous solution (22.5 μL) were added to the suspension. After mixing with ultrasonication for 5 min, the suspension was stirred at elevated temperature (~100 °C) using a glass rod until the solution was completely evaporated. RhCrO$_x$ (imp) loaded SrTiO$_3$:Al was obtained after calcining at 350 °C for 1 h with a ramping rate of 10 °C min$^{-1}$.

SrTiO$_3$:Al loaded with Rh, Cr$_2$O$_3$, and CoOOH was also prepared by photodeposition according to a reported method[19]. The SrTiO$_3$:Al (100 mg) was dispersed in ultrapure water (75 mL), and a 0.01 M RhCl$_3$ aqueous solution (100 μL) prepared from RhCl$_3$ 6H$_2$O (FUJIFILM Wako Pure Chemical Corporation) was dropped into the suspension. The suspension was then irradiated with 370-nm LED light (Asahi Spectra Co., Ltd.; CL-1501; details are shown in Supplementary Fig. 2) for 10 min with stirring. Subsequently, 0.01 M K$_2$CrO$_4$ (Kanto Chemical Co. Ltd.) aqueous solution (50 μL) was added and then irradiated with LED light

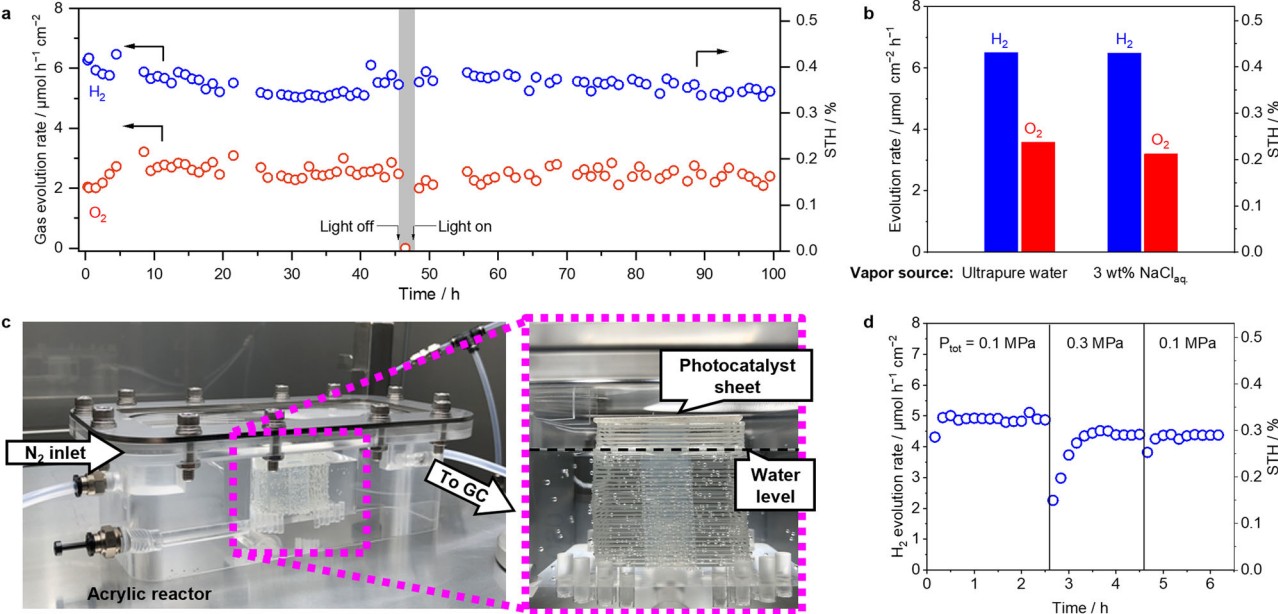

**Fig. 5 | Practical feasibility study of photocatalytic overall water splitting under vapor feeding. a** Durability test of CoOOH/Rh loaded SrTiO$_3$:Al coated with TiO$_x$ under saturated water vapor feeding balanced with Ar at 24 °C (10 mL min$^{-1}$, $p_{H2O}$ = 2.9 kPa). The light source was a simulated sunlight (AM 1.5G). **b** The overall water splitting rate of CoOOH/Rh loaded SrTiO$_3$:Al coated with TiO$_x$ under water vapor feeding supplied from ultrapure water or brine (3 wt% NaCl$_{aq}$) balanced with Ar at 24 °C. **c** Photographic images of a vaper feeding photocatalytic overall water splitting system under pressurized condition. **d** H$_2$ evolution rate of CoOOH/Rh loaded SrTiO$_3$:Al coated with TiO$_x$ under pressurized condition at total pressure ($P_{tot}$) of 0.1 or 0.3 MPa using the system shown in panel **c**. The light source was simulated sunlight (AM 1.5G).

as above for 5 min followed by the addition of 85 mM $Co(NO_3)_2$ aqueous solution (10 μL) and another 5-min irradiation. The solvent was not exchanged during this multistep photodeposition.

## $TiO_x$ or $TaO_x$ coating on $SrTiO_3$:Al photocatalysts

$TiO_x$ or $TaO_x$ was coated on the $SrTiO_3$:Al loaded with Rh, $Cr_2O_3$, and CoOOH via photodeposition in the same solvent used for Rh, $Cr_2O_3$, and CoOOH deposition before the filtration[26]. For $TiO_x$ coating, titanium tetraisopropoxide (FUJIFILM Wako Pure Chemical Corporation, 11.1 μL, 37 μmol) with 30 vol% $H_2O_2$ (FUJIFILM Wako Pure Chemical Corporation, 200 μL, 3.2 mmol) was added into the suspension containing $SrTiO_3$:Al loaded with Rh, $Cr_2O_3$, and CoOOH. This was then mixed via ultrasonication for 5 min. Subsequently, 1 M NaOH aqueous solution (74.6 μL) was added to the solution where the ratio between the added $Na^+$ versus Ti contained in titanium tetraisopropoxide was 2:1. For $TaO_x$ coating, tantalum(V) chloride (MilliporeSigma, 8.0 mg, 22 μmol) with 30 vol% $H_2O_2$ (FUJIFILM Wako Pure Chemical Corporation, 500 μL, 8.0 μmol) was added to the suspension containing the $SrTiO_3$:Al loaded with Rh, $Cr_2O_3$, and CoOOH. This was then mixed with ultrasonication for 5 min. Subsequently, 1 M NaOH aqueous solution (113 μL) was added to the solution where the ratio between the added $Na^+$ versus Ta contained in tantalum(V) chloride was 5:1. The resulting suspension was irradiated with 370 nm LED (730 mW) for 12 h, and then the precipitates were separated by filtration and washed with ethanol three times. The collected slurry was not dried further.

## Characterization

X-ray diffraction (XRD) patterns were collected using RINT-UltimaIII (Rigaku Corporation) with a Cu Kα X-ray radiation source ($\lambda$ = 0.154056 nm) at a scan rate of 30° min$^{-1}$. The X-ray photoelectron spectroscopy (XPS) measurements were performed on PHI5000 VersaProbe (ULVAC Inc.) with an Al−Kα source operating at 5 kV. STEM images were collected using JEM-2800 transmission electron microscopes (JEOL Ltd.) or Talos 200kV-FE-(S)TEM with super X XDS (Thermo Fisher Scientific Inc.) operating at 200 kV in conjunction with EDS using an X-MAX 100TLE SDD detector (Oxford Instruments). Water adsorption tests and $N_2$ adsorption tests were performed on a 3Flex gas adsorption measurement device (Micromeritics Instrument Corp.). The samples were pretreated at 60 °C for 12 h under vacuum. The adsorbed gas amount was measured at the equilibrium point where the pressure change becomes less than 0.01% within 30 s.

Diffuse reflectance infrared Fourier transform spectra (DRIFTS) were recorded on an FT/IR-6600 (JASCO Corporation) equipped with a heat-vacuum diffuse reflection cell (S.T. Japan Inc., Heat Chamber Type-1000 °C). Pelletized powder samples were placed at the focal point of an integrating sphere. After drying at 60 °C for 1 h under vacuum conditions, the samples were cooled to room temperature under dry Ar (20 mL min$^{-1}$), and baseline collection was then performed. Subsequently, Ar with saturated $H_2O$ or $D_2O$ vapor (20 mL min$^{-1}$) was fed into the cell for 30 min, and the spectrum was then recorded. UV–Vis diffuse reflectance spectrum of samples was recorded using a V-770 UV-Vis spectrometer (JASCO Corporation) equipped with an integrating sphere (JASCO Corporation, ISN-923) and a spectralon block as a reference to adjust the 100% reflectance level and set the reflectance of the sample holder to 0%.

## Photocatalytic reactions

Photocatalytic reactions were carried out in the continuous gas flow reactor (Supplementary Fig. 1a). Each synthesized catalyst powder was dispersed in a tiny amount of ultrapure water, dropped onto a frosted glass plate (SCHOTT Corp. Tempax), and then dried at 60 °C for 10 min using a hotplate. The loaded amount of the photocatalyst was 5 mg per 1 cm$^2$ of the glass plate. The prepared photocatalyst sheet was placed at the bottom of a cylindrical reactor made of stainless steel with an internal volume of 38 cm$^3$. Water vapor was

supplied to the reactor with Ar gas through a water bubbler, which was filled with ultrapure water or brine (3 wt% NaCl aqueous solution). In the pipeline between the water bubbler and the reactor, a hygrothermograph (Bosch BME280) was inserted.

For the activity test under controlled temperature and relative humidity, the photocatalyst reactor, the hygrothermograph, and the water bubbler were placed in a heating oven, and the relative humidity of the supplied water vapor was controlled by changing the ratio of the flow rate of the dry Ar and the wet Ar, which contained saturated water vapor. To estimate the effect of $O_2$ addition in inlet gas, a $O_2$/Ar mixture with a defined ratio was introduced into the water bubbler and then supplied to the reactor. The total flow rate was 10 to 50 mL min$^1$.

The photocatalyst sheet was irradiated with a 370-nm LED (Asahi Spectra Co., Ltd.; CL-1501) equipped with rod lens (RLQL80-05) to provide a square-shaped light with uniform intensity or simulated sunlight (Asahi Spectra Co.; AM1.5 G, solar simulator HAL-320) through a window on top of the reactor. The amount of generated $H_2$ and $O_2$ gases were measured using gas chromatography (GC-8A; Shimadzu Co.) equipped with a 5 Å molecular sieve column and a thermal conductivity detector.

The apparent quantum yield (AQY) for overall water splitting under LED excitation with a center wavelength of 370 nm was calculated according to the following equation:

$$AQY(\%) = \frac{2 \times r(H_2)}{r(photons)} \times 100 \qquad (1)$$

Here, $r(H_2)$ and $r$(photons) represent the $H_2$ evolution rat9e and the number of the photons reaching the surface of the photocatalyst sheet per unit time, respectively. The photons in the incident light with wavelengths longer than the absorption edge of $SrTiO_3$:Al−as determined by the UV-Vis diffuse reflectance spectrum (Supplementary Fig. 4)−were excluded from the AQY calculation.

The solar to hydrogen (STH) efficiency for overall water splitting under simulated sunlight was determined according to the following equation.

$$STH(\%) = \frac{r(H_2) \times \triangle G(H_2O)}{I \times S} \times 100 \qquad (2)$$

Here, $\Delta G(H_2O)$, $I$, and $S$ denote the reaction Gibbs energy of the water splitting reaction $H_2O(l) \rightarrow H_2(g) + 1/2O_2(g)$ (237 kJ mol$^{-1}$ at 288 K), the energy intensity of AM 1.5G solar irradiation (100 mW cm$^{-2}$), and the photocatalyst sheet area (4 cm$^2$), respectively.

## in situ X-ray absorption measurement

X-ray absorption fine structure (XAFS) of Ta L$_{III}$ edge in $TaO_x$ coated $SrTiO_3$:Al loaded with CoOOH/Rh was measured at BL01B1 beamline at SPring-8 synchrotron facility (Hyogo, Japan) using a ring energy of 8 GeV (proposal No. 2021B1610). The incident X-ray was monochromated using Si(111) double crystal, and the energy was calibrated using a Ta foil. A pelletized sample ($\varphi$7mm) of $TaO_x$ coated $SrTiO_3$:Al loaded with CoOOH/Rh was placed in a quartz cell with polyimide windows connected to a $N_2$ gas cylinder. After pretreating under dry $N_2$ flow (100 mL min$^{-1}$) for 30 min, the sample was cooled to room temperature, and the spectrum was then collected. $N_2$ containing saturated water vapor was then introduced. Spectral measurements were then performed after 30 min. The sample was then further treated at 250 °C under dry $N_2$ flow followed by spectral measurements under dry and wet $N_2$ flow at room temperature. For EXAFS analysis, a Fourier transformation of the EXAFS oscillation was performed between 3 and 14 Å$^{-1}$ with k-weight of 2. Peak fittings of the EXAFS results were performed using FEFF code. Debye−Waller factor ($\sigma^2$) and the interatomic distance of Ta and O ($R$) were calculated from the fitting.

## Photocatalytic reactions at pressurized condition

A Taylor-made acrylic reactor with an internal volume of 200 cm$^3$ (Fig. 5c) was used to demonstrate the overall water splitting under water vapor feeding at pressurized conditions. The reactor was filled with ultrapure water, and the glass pedestal floated in the water. The photocatalyst sheet (4 × 4 cm$^2$) immobilized a TiO$_x$ coating on the SrTiO$_3$:Al loaded with Rh and CoOOH; this was placed on the pedestal, i.e., the photocatalyst sheet was placed above the water level. The inlet N$_2$ gas flow (50 mL min$^1$) was introduced into the gas phase inside the acrylic(reactor through the filled liquid water. Therefore, the relative humidity in the gas phase inside the acrylic reactor was almost unity. The total pressure in the gas phase inside the acrylic reactor was controlled from 0.1 to 0.3 MPa using a back pressure regulator placed downstream of the reactor. The photocatalyst sheet was irradiated with simulated sunlight (AM 1.5G) from the top. The amount of generated H$_2$ was measured by gas chromatography (Agilent 490 Micro GC).

## Data availability

All data supporting the findings of this study are available within this article and its Supplementary Information. Source data that support the findings of this study are available from the corresponding author upon reasonable request. Correspondence and requests for materials should be addressed to Takanabe, K.

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

## Acknowledgements

This work is financially supported by the Artificial Photosynthesis Project (ARPChem) of the New Energy and Industrial Technology Development Organization (NEDO), and the Mohammed bin Salman Center for Future Science and Technology for Saudi-Japan Vision 2030 at The University of Tokyo (MbSC2030). We would like to thank Advanced Characterization Nanotechnology Platform in the Nanotechnology Platform Project sponsored By the Ministry of Education,

Culture, Sports, Science and Technology (MEXT), Japan. (JPMXP09-A-21-UT-0046) for TEM observations.

## Author contributions

K.T., F.K., and T.S. conceived the project idea, designed the research, and co-wrote the manuscript. T.S. carried out the materials synthesis and all the experiments except for TEM measurements and a vaper feeding photocatalytic overall water splitting test under pressurized condition. N.K. and T.F. conducted a vaper feeding photocatalytic overall water splitting test under pressurized condition and provided the TEM image of CoOOH/Rh loaded SrTiO$_3$:Al coated with TaO$_x$. M.N. and N.S. contributed the TEM measurement for CoOOH/Rh loaded SrTiO$_3$:Al coated with TiO$_x$. T.T. and K.D. contributed to the synthesis of the materials and validity of experimental results. All authors contributed to the discussion of results and commented on the manuscript.

## Competing interests

The authors declare no competing interests.
