## [Peer Review File · Nature Communications]

A hygroscopic nano-membrane coating achieves efficient vapor-fed photocatalytic water splittingREVIEWER COMMENTS

Reviewer #1 (Remarks to the Author):

In this manuscript, Suguro et al presents the coating of thin layer (of few nm in thickness) of amorphous TiO_x or TaO_x on the surface of SrTiO₃ nanoparticles that shows enhanced performance for the solar water splitting using H₂O vapor. Indeed, TiO_x or TaO_x layer increase the water adsorption/condensation of the nanoparticle photocatalyst. Thus, SrTiO₃ nanoparticles photocatalysts could operate under water vapor as it do in liquid water. For the 100h-photocatalytic experiment under the simulated sunlight, a modest Solar-to-H₂ conversion yield of 0.4% was achieved.

In general, the work presents a nice set of experiment data which could be of interests for those who work in the field of artificial photosynthesis. However, I am not convinced about the potential impact of this work, e.g. how other scientists could learn from this approach to further develop photocatalytic vapor water splitting. Actually, progress presented in this work is not significant compared with those published by the same group for the system SrTiO₃.

I also note that the current work focuses mainly on the fabrication of materials and its photocatalytic performance assaying rather than deeply investigating the actual role of TiO_x, TaO_x layer. Several discussion remains as speculation without any direct evidences, e.g. lines 95-97 speculating on the reason behind the decrease of H₂ evolution rate under high light irradiation intensity (>59 mW cm⁻²), lines 203-204 mentioning about the irreversible degradation of TaO_x coating layer.

To conclude, this work could be publishable in a more specific journal but not for the Nature Communication.

Reviewer #2 (Remarks to the Author):

The authors proposed a photocatalytic system where the presence of a surface TiO_x (or TaO_x) layer improves significantly the performance under water vapor feeds for photocatalytic water splitting, bridging the existing gap in observed AQY between liquid water experiments and previous water vapor experiments. These results and the proposed approach are a breakthrough in photocatalytic water splitting by water vapor splitting, and will certainly open new possibilities for investigations in this direction. The materials are well characterized and the catalytic performances reported not only under UV but also under AM1.5G simulated solar irradiation, including 100 h stability measurements.

For the mentioned reason, I recommend the publication in this journal. I have only few comments, that I recommend the Authors to address before publication.

General Comment:

Page 9, line 143 and Figure S11: Figure S11 seems to indicate that the TiO_x layer covers (or at least partially covers) also the Rh nanoparticle co-catalysts. Does it cover also the CoOOH co-catalysts? Due to the complex photocatalytic system design (photocatalyst, multiple co-catalysts, TiO_x layer), a drawing would be useful. For example, as insert of Figure 1a.

Minor comments

Page 2, line 42: "The STH efficiency of 10% has long durability and is needed for commercial feasibility". The first time that I have read this sentence it was not clear to me if it is meant to be a general sentence describing what a photocatalyst should achieve or if the authors were referring to a previously discussed value in the text, which I couldn't find. A STH of 10% was measured in this work under certain conditions, but it is only discussed later, and in the sentence before the verb that refers to the experiments is in the past tense is used but here the present tense is used. The result of STH of 10% can be anticipated here, but I suggest to rewrite this sentence.

Page 13, Figure 3e: the panel shows 4 curves, which are well described in the caption. However, the panel does not have a legend. I believe that a compact legend will help the reader, for example “—dry, After 60°C”, “vapor, after 60°C”, “dry, after 250°C”, ... Or similar can be added in the top left corner.

Reviewer #3 (Remarks to the Author):

This work describes an innovative approach for highly efficient water splitting using water vapor (rather than liquid), which works by depositing a nano-membrane of TiOx or TaOx onto Al-doped SrTiO3. A high efficiency (AQY) of ~54% is achieved (comparable to the best liquid-based photocatalysis) that represents a breakthrough and is an exciting new advance in this field. The main novelty is in the hydrophilic TiOx/TaOx layers, that are both permeable and help to form a layer of water at the particles' surfaces. Even more, the process is also shown to work equally well with seawater and has significant advantages for scaled up application. Thus, I am recommending publication of this manuscript after a revision that can address the below suggestions and comments.

a) On page 4, I would suggest changing “there are no losses in these elementary steps” to “there are no losses in the first 4 elementary steps”

b) On the bottom of page 5, the authors compare the H2 evolution rate with the calculated rate when the STH efficiency reaches ~10% under AM1.5G. As the H2 rate was measured under irradiation with a 370 nm LED, it is unclear how this rate can be compared to a rate under the full solar spectrum?

c) The authors imply that the TiOx layer prevents the reverse redox reaction. The authors should try and clarify how this might be possible.

d) The authors describe that the TaOx coating becomes more dense after heat treatment and dehydration, with a change in the Ta-O radial distance from 1.44 to 1.41 Angstroms. Can the standard deviations be provided on these distances to determine whether they are statistically different?

e) What type of frosted glass plate was used and what is its commercial source?

Reviewer 1

In this manuscript, Suguro et al presents the coating of thin layer (of few nm in thickness) of amorphous TiOx or TaOx on the surface of SrTiO₃ nanoparticles that shows enhanced performance for the solar water splitting using H₂O vapor. Indeed, TiOx or TaOx layer increase the water adsorption/ condensation of the nanoparticle photocatalyst. Thus, SrTiO₃ nanoparticles photocatalysts could operate under water vapor as it do in liquid water. For the 100h-photocatalytic experiment under the simulated sunlight, a modest Solar-to-H₂ conversion yield of 0.4% was achieved.

We would like to thank Reviewer #1 for his/her review. We have addressed the point raised by the reviewer below and believe that our modifications mitigate the concerns of the reviewer.

1. In general, the work presents a nice set of experiment data which could be of interests for those who work in the field of artificial photosynthesis. However, I am not convinced about the potential impact of this work, e.g. how other scientists could learn from this approach to further develop photocatalytic vapor water splitting. Actually, progress presented in this work is not significant compared with those published by the same group for the system SrTiO₃.

For practical use, water vapor splitting systems can mitigate the following issues of the conventional liquid phase system¹;

- Catalyst components may leach into the water for long run
- Bubbles generated at the surface of the photocatalyst can attach and reduce the water splitting rate due to light scattering and backward reactions.
- In the unlikely event of water freezing in the photoreactor, the reaction stops until the ice melts and freezing damages the photoreactor system.
- Need for a sturdy container that can withstand the weight and viscosity of water and high cost of pumps to transport the water.

However, the water vapor splitting system has not been intensely investigated due to the low efficiencies.

Our findings for highly efficient overall water splitting under vapor feeding enable not only the development of efficient photocatalysts for vapor splitting, but also a paradigm shift in the reactor engineering for water splitting systems. We believe that this manuscript will be of interest to a large chemistry audience, especially in catalysts and chemical engineering.

We revised the manuscript by adding the following sentences.

line 46-48:

“Although water vapor splitting systems can mitigate the above mentioned issues of the conventional liquid phase system, the water vapor splitting system has not been intensely investigated because of lack of strategies to archive high efficiencies¹.”

We also added the following reference as No.13.

[1] Suguro, T., Kishimoto, F., Takanabe, K. Photocatalytic Hydrogen Production under Water Vapor Feeding - A minireview. *Energy & Fuels* doi:10.1021/acs.energyfuels.2c01478.

2. I also note that the current work focuses mainly on the fabrication of materials and its photocatalytic performance assaying rather than deeply investigating the actual role of TiO_x, TaO_x layer. Several discussion remains as speculation without any direct evidences, e.g. lines 95-97 speculating on the reason behind the decrease of H₂ evolution rate under high light irradiation intensity (>59 mW cm⁻²), lines 203-204 mentioning about the irreversible degradation of TaO_x coating layer.

According to the comment on lines 95-97, we considered that the reason for the decrease of H₂ evolution rate under high light irradiation should be discussed in the characterization of oxide thin layer section. We additionally conducted infrared diffuse reflectance spectroscopy to follow OH vibration region before and after the 370 nm LED irradiation under saturated water vapor (Figure R1). The difference spectra indicate a decrease in the number of OH species with increasing the incident light intensity, according to the loss of weight loss observed in the TGA measurement.

We may argue that the irreversible deactivation behavior was clearly evident by photocatalytic performance we have already demonstrated. Combining H₂ evolution test, in situ XAFS measurement and TG analysis, these results suggested that an irreversible structural change of TaO_x layer was induced by desorption of water molecules and hydroxyl groups through thermal treatment, resulting in loss of water adsorption capacity.

Figure R1: Difference spectra of diffuse reflectance of TiO_x coated, CoOOH/Rh loaded SrTiO₃:Al in OH vibration before and after 20 min light irradiation. The feed gas was Ar with saturated water vapor at 24 °C. Incident light: 370 nm LED.

We revised the manuscript by adding the following sentences.

line 98-100:

“The cause of decrease in AQY as the function of light intensity as well as decrease in formation rate with time will be discussed in the characterization of oxide thin layer section.”

and line 199-207:

“TG–DTA measurement (Figure S19 and S20) also shows that the weight loss by dehydration occurs starting at 45 °C, consistent with the structural change measured by EXAFS. These results suggested that an irreversible structural change of TiO_x or TaO_x layer was induced by loss of water molecules from the layer, resulting in reduced water adsorption and lower H₂ production rate. The decrease in AQY as increasing light intensity and the reaction time shown in Figure 2(d) can be due to the loss of water molecules in the TiO_x or TaO_x layer either by temperature rise or higher consumption rate of water molecules than their supply by water splitting reaction. Diffuse reflectance infrared spectroscopy measurement as shown in Figure S21 supports the loss of OH species from TiO_x layer under the 370 nm LED irradiation with high light intensity.”

Figure R1 was added as Figure S21.

Reviewer 2

The authors proposed a photocatalytic system where the presence of a surface TiO_x (or TaO_x) layer improves significantly the performance under water vapor feeds for photocatalytic water splitting, bridging the existing gap in observed AQY between liquid water experiments and previous water vapor experiments. These results and the proposed approach are a breakthrough in photocatalytic water splitting by water vapor splitting, and will certainly open new possibilities for investigations in this direction. The materials are well characterized and the catalytic performances reported not only under UV but also under AM1.5G simulated solar irradiation, including 100 h stability measurements. For the mentioned reason, I recommend the publication in this journal. I have only few comments, that I recommend the Authors to address before publication.

We would like to thank Reviewer #2 for his/her review. We have addressed the point raised by the reviewer below and believe that our modifications mitigate the concerns of the reviewer.

General Comment:

1. Page 9, line 143 and Figure S11: Figure S11 seems to indicate that the TiO_x layer covers (or at least partially covers) also the Rh nanoparticle co-catalysts. Does it cover also the CoOOH co-catalysts? Due to the complex photocatalytic system design (photocatalyst, multiple co-catalysts, TiO_x layer), a drawing would be useful. For example, as insert of Figure 1a.

Although CoOOH species were not clearly observed in TEM images due to the trace amount of Co deposited (0.05 wt% of Co), it can be expected that TiO_x should cover the highly dispersed CoOOH.

According to the reviewers suggestion, we added the following schematic illustration of the fabricated catalyst (Figure R2) as Figure 1, where CoOOH/Rh loaded SrTiO₃:Al is uniformly coated with amorphous oxide (TiO_x or TaO_x) thin layer.

Figure R2. Schematic image of amorphous metal oxide coated, CoOOH/Rh loaded SrTiO₃:Al.

Minor comments:

1. Page 2, line 42: "The STH efficiency of 10% has long durability and is needed for commercial feasibility". The first time that I have read this sentence it was not clear to me if it is meant to be a general sentence describing what a photocatalyst should achieve or if the authors were referring to a previously discussed value in the text, which I couldn't find. A STH of 10% was measured in this work under certain conditions, but it is only discussed later, and in the sentence before the verb that refers to the experiments is in the past tense is used but here the present tense is used. The result of STH of 10% can be anticipated here, but I suggest to rewrite this sentence.

According to the comment, the authors revised the sentence in lines 39-40 (line 42 in the previous manuscript) as "The STH efficiency of 10% is often the benchmark target for green water splitting technology for commercial feasibility".

2. Page 13, Figure 3e: the panel shows 4 curves, which are well described in the caption. However, the panel does not have a legend. I believe that a compact legend will help the reader, for example "—dry, After 60°C", "vapor, after 60°C", "dry, after 250°C", ... Or similar can be added in the top left corner.

According to the comment, we added the legend in Figure 4e (Figure 3e in the previous manuscript) as shown in Figure R3.

Figure R3. Fourier transformed Ta L_{III} EXAFS patterns of CoO_x/Rh loaded $SrTiO_3:Al$ coated with TaO_x recorded at 24 °C under dried condition (dashed) and (solid) after thermal treatment.

Reviewer 3

This work describes an innovative approach for highly efficient water splitting using water vapor (rather than liquid), which works by depositing a nano-membrane of TiO_x or TaO_x onto Al-doped SrTiO₃. A high efficiency (AQY) of ~54% is achieved (comparable to the best liquid-based photocatalysis) that represents a breakthrough and is an exciting new advance in this field. The main novelty is in the hydrophilic TiO_x/TaO_x layers, that are both permeable and help to form a layer of water at the particles' surfaces. Even more, the process is also shown to work equally well with seawater and has significant advantages for scaled up application. Thus, I am recommending publication of this manuscript after a revision that can address the below suggestions and comments.

We would like to thank Reviewer #3 for his/her review and constructive comments. We have addressed the point raised by the reviewer below and believe that our modifications mitigate the concerns of the reviewer.

1. On page 4, I would suggest changing “there are no losses in these elementary steps” to “there are no losses in the first 4 elementary steps”

We agree with the reviewer. We have revised the sentence as suggested (line 60).

2. On the bottom of page 5, the authors compare the H₂ evolution rate with the calculated rate when the STH efficiency reaches ~10% under AM1.5G. As the H₂ rate was measured under irradiation with a 370 nm LED, it is unclear how this rate can be compared to a rate under the full solar spectrum?

The water splitting test under high light irradiation (Figure 2(b)) shows that the H₂ evolution rate reached to ~150 μmol h⁻¹ per 1 cm² photocatalyst sheet, demonstrating the applicability of the TiO_x coating for improved photocatalysts where solar-to-hydrogen (STH) efficiency of 10 % (corresponding to a hydrogen production rate of 150 μmol h⁻¹ per 1 cm²) is achievable. This demonstration suggested that this H₂ production system is not limited by water vapor supply even if the H₂ evolution rate is corresponding to the rate of 10% STH.

The above sentence was added in lines 102-106:

“Figure 2(b) also shows that the H₂ evolution rate reached to ~150 μmol h⁻¹ per 1 cm² photocatalyst sheet, demonstrating the applicability of the TiO_x coating for improved photocatalysts where solar-to-hydrogen (STH) efficiency of 10 % (corresponding to a hydrogen production rate of 150 μmol h⁻¹ per 1 cm²) is achievable. This demonstration suggested that this H₂ production system is not limited by water vapor

supply even if the H₂ evolution rate is corresponding to the rate of 10% STH.”

3. The authors imply that the TiO_x layer prevents the reverse redox reaction. The authors should try and clarify how this might be possible.

According to the reviewer’s comment, we have carefully revised the discussion of TiO_x layer for preventing the reverse redox reaction.

It has been previously demonstrated that TiO_x coated on SrTiO₃ based photocatalyst can prevent the reverse redox reaction in liquid water as an H₂-selective nanomembrane (ref. 6). However, there was no report under water vapor feeding. As shown in Figure R4, H₂ evolution rate slightly decreased with increasing the partial pressure of intentionally introduced O₂ in the fed gas; the apparent H₂ evolution rate was already almost zero in the presence of 4% O₂ using CoOOH/Rh loaded SrTiO₃:Al. Thus, we considered that TiO_x layer considerably prevents the reverse water-forming reaction under water vapor feeding.

Fig. R4: Dependence of H₂ evolution rate of CoOOH/Rh loaded SrTiO₃:Al coated with TiO_x (blue) and without TiO_x (green) on O₂ partial pressure ranged from 0 to 33 kPa under 370 nm LED (5.1 mW cm⁻²).

We revised the discussion in lines 118-130:

“The TiO_x coated CoOOH/Rh loaded SrTiO₃:Al photocatalyst exhibited a AQY comparable to that of TiO_x coated CoOOH/Cr₂O₃/Rh loaded SrTiO₃:Al photocatalyst.

The Cr₂O₃ co-loaded with Rh co-catalyst has been reported for improved overall apparent quantum yield of water splitting in liquid water by preventing the reverse reaction of water generation from H₂ and O₂ (ref. 25, 27). However, because of the possibility of harmful Cr leaching, detailed characteristics of Cr-free TiO_x coated CoOOH/Rh loaded SrTiO₃:Al photocatalyst were investigated. The AQY of water splitting under vapor feeding using the photocatalysts with or without Cr₂O₃ was almost unchanged as shown in Figure 2(c). Indeed, it has been previously demonstrated that TiO_x layer acting as an H₂-selective nanomembrane can prevent the reverse water-forming reaction in liquid water (ref. 6). To demonstrate the dynamics of reverse water-forming reaction from H₂ and O₂, photocatalytic performance in the presence of O₂ was tested. The results are shown in Figure 2(d). The apparent H₂ evolution rate was already almost zero in the presence of 4% O₂ using CoOOH/Rh loaded SrTiO₃:Al photocatalyst without TiO_x layer; the apparent H₂ evolution rate linearly slightly decreased as a function of O₂ partial pressure using CoOOH/Rh loaded SrTiO₃:Al photocatalyst with TiO_x layer, but the AQY was maintained above 40% even in the presence of 20% O₂ corresponding O₂ partial pressure in the ambient air. TiO_x layer considerably prevents the reverse water-forming reaction under water vapor feeding.”

Figure 2(d) (Figure 1(d) in the previous manuscript) was changed to Figure R4.

4. The authors describe that the TaO_x coating becomes more dense after heat treatment and dehydration, with a change in the Ta-O radial distance from 1.44 to 1.41 Angstroms. Can the standard deviations be provided on these distances to determine whether they are statistically different?

This comment pointed out the certainty of the structural changes demonstrated by the XAFS results. The direct evidence of the dehydrogenative transformation of TaO_x layer after drying at 250 °C was given by TG-DTA results as shown in Figure S19.

XAFS results provide the local structure of the photocatalysts. While the reviewer suggested to calculate the standard deviations of EXAFS, estimating standard deviations based on multiple measurements is not realistic due to limited beam time for XAFS measurement. Therefore, we have performed detailed analysis of the EXAFS profiles using FEFF code. The fitting results are shown in Figure R5. The neighboring oxygen shell were deconvoluted into two peaks assigned as Ta-O₁ and Ta-O₂ based on the crystal structure of orthorhombic Ta₂O₅. Table R1 summarizes the fitting results of Debye-Waller factor (σ^2) and the interatomic distance (R). Because

these measurements were performed at constant temperature, Debye-Waller factor (σ^2) corresponds to the statistical degree of disorder in atomic positions with respect to the ideal crystal structure of Ta₂O₅. Accompanied with the decrease of σ^2 of both Ta-O₁ and Ta-O₂ peaks after the treatment at 250 °C, the interatomic distance (R) of Ta and O became slightly shorter. These results suggested that, as a statistically significant difference, TaO_x layer should become denser and comparatively ordered crystal structure after the treatment at 250 °C even though there is no significant change in the X-ray absorption near edge structure (XANES) region before and after heat treatment (Figure 4(f)).

Figure R5. FEFF fitting of Ta L_{III} edge EXAFS patterns of CoOOH/Rh loaded SrTiO₃:Al coated with TaO_x recorded at 24 °C under saturated water vapor feeding after heat treatment after heat treatment at 60 °C (a) and 250 °C (b). The top lines are Fourier transformed patterns. The bottom lines are the patterns before the Fourier transform. Experimental results (blue) were fitted by FEFF (red). The radial distance range for fittings was 1–2.5 (green).

Table R1. Debye-Waller factor (σ^2) and interatomic distance (R) from FEFF fitting of EXAFS shown in Figure R2, where Ta-O₁ represents scattering concerning the nearest O₂ and Ta-O₂ concerning the second nearest O₂. Orthorhombic Ta₂O₅ crystal structure (Pmmm) was used as reference structure.

	σ^2	R
Under vapor, after drying at 60 °C	Ta-O ₁	0.011 1.91
	Ta-O ₂	0.0059 2.12
Under vapor, after drying at 250 °C	Ta-O ₁	0.0087 1.87
	Ta-O ₂	0.0036 2.11

We added the above discussion on line 199-207 as below:

“The peaks at 1.4 Å in the radial distribution functions, attributed to the neighboring oxygen shell (ref. 28), were deconvoluted into two peaks assigned as Ta-O₁ and Ta-O₂ based on the crystal structure of orthorhombic Ta₂O₅ (Figure S18). Table 1 summarizes the fitting results of Debye-Waller factor (σ^2) and the interatomic distance (R). Because these measurements were performed at constant temperature, Debye-Waller factor (σ^2) corresponds to the statistical degree of disorder in atomic positions with respect to the ideal crystal structure of Ta₂O₅. Accompanied with the decrease of σ^2 of both Ta-O₁ and Ta-O₂ peaks after the treatment at 250 °C, the interatomic distance (R) of Ta and O became slightly shorter. These results suggested that, as a statistically significant difference, TaO_x layer should become denser and comparatively ordered crystal structure after the treatment at 250 °C even though there is no significant change in the X-ray absorption near edge structure (XANES) region before and after heat treatment (Figure 4(f)).”

We also added the below sentences to lines 382-383 in “Method” section:

“Peak fittings of the EXAFS results were performed using FEFF code. Debye-Waller factor (σ^2) and the interatomic distance of Ta and O (R) were calculated from the fitting.”

The location of the XANES profiles was changed from supporting information to Figure 4(f). Figure R5 and Table R1 were added as Figure S18 and Table 1, respectively.

5. What type of frosted glass plate was used and what is its commercial source?

We added the information of the frosted glass plate as “frosted glass plate (SCHOTT Corp. Tempax)” in line 344.

REVIEWERS' COMMENTS

Reviewer #2 (Remarks to the Author):

The authors addressed my comments and included a graphical cartoon of the photocatalytic system as suggested. I recommend publication in this Journal.

Reviewer #3 (Remarks to the Author):

This is a review of a revised version of the manuscript. The authors have suitably addressed all of my previous comments and questions. I am therefore recommending publication of the manuscript in its current form.